# DIVERSITY OF GENERATED UNLABELED DATA MATTERS FOR FEW-SHOT HYPOTHESIS ADAPTATION

## ABSTRACT

Generating unlabeled data has been recently shown to help address the *few-shot hypothesis adaptation* (FHA) problem, where we aim to train a classifier for the target domain with a few labeled target-domain data and a well-trained source-domain classifier (i.e., a source hypothesis), for the additional information of the highly-compatible unlabeled data. However, the generated data of the existing methods are extremely similar or even the same. The strong dependency among the generated data will lead the learning to fail. In this paper, we propose a *diversity-enhancing generative network* (DEG-Net) for the FHA problem, which can generate diverse unlabeled data with the help of a kernel independence measure: the *Hilbert-Schmidt independence criterion* (HSIC). Specifically, DEG-Net will generate data via minimizing the HSIC value (i.e., *maximizing the independence*) among the semantic features of the generated data. By DEG-Net, the generated unlabeled data are more diverse and more effective for addressing the FHA problem. Experimental results show that the DEG-Net outperforms existing FHA baselines and further verifies that generating diverse data plays an important role in addressing the FHA problem.

## 1 INTRODUCTION

Data and expert knowledge are always scarce in newly-emerging fields, thus it is very important and challenging to study how to leverage knowledge from other similar fields to help complete tasks in the new fields. To cope with this challenge, transfer learning methods were proposed to leverage knowledge of source domains (e.g., data in source domains or models trained with data in source domains) to help complete the tasks in other similar domains (a.k.a. the target domains) (Fang et al., 2020; Jing et al., 2020; Pan & Yang, 2009; Sun et al., 2019; Teshima et al., 2020; Zamir et al., 2018). Among many transfer learning methods, *hypothesis transfer learning* (HTL) methods have received a lot of attention since it does not require access to the data in source domains, which prevents the data leakage issue and protects the data privacy (Chi et al., 2021a; Du et al., 2017; Liang et al., 2020; Yang et al., 2021a;b). Recently, the *few-shot hypothesis adaptation* (FHA) problem has been formulated to make HTL more realistic, which is suitable to solve many problems (Liu et al., 2021; Snell et al., 2017; Wang et al., 2020; Yang et al., 2020). In FHA, only a well-trained source-domain classifier (i.e., source hypothesis) and few labeled target-domain data are available (Chi et al., 2021a).

Similar to HTL, FHA also aims to obtain a good target-domain classifier with the help of a source hypothesis and few target-domain data (Chi et al., 2021a; Motiian et al., 2017). Recently, *generating unlabeled data* has been shown to be an effective strategy to address FHA (Chi et al., 2021a). The *target-oriented hypothesis adaptation network* (TOHAN), a one-step solution to the FHA problem, constructed an intermediate domain to enrich the training data. The data in the intermediate domain are highly compatible with both source domain and target domain (Balcan & Blum, 2010). By the generated unlabeled data in the intermediate domain, TOHAN partially overcame the problems caused by data scarce in the target domain.

However, the existing methods *ignore* the diversity of the generated data or the independence among the generated data, so that the generated data are extremely similar or even the same. Lack of diversity leads to less effective data for addressing the FHA problem. Taking the FHA task of digits datasets as an example, we found that the data generated by TOHAN has an issue that the generator tends to *copy* target data (Figure 1(a)). To show how *diversity* matters in the FHA problem, we conduct the

target data   generated data


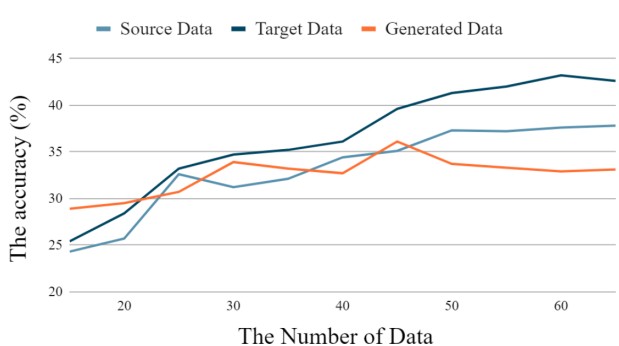

(a) The copy issue of generated data on $M \to S$ task

(b) Classification accuracy of different amount of unlabeled data drawn from different domains on $M \to S$ task

Figure 1: The low-diversity issue of generated unlabeled data when solving the FHA problem. Subfigure (a) illustrates the labeled data (left) drawn from the target domain and unlabeled data (right) generated by TOHAN on the MNIST→SVHN ($M \to S$) task. It is clear that the generated data are similar to each other and seem to copy the original target data (middle, left). Subfigure (b) illustrates the accuracy of the data drawn from different domains with different data volumes on the task $M \to S$. For the source data and target data, the accuracy of the model trained by them is higher as the number of the data increases. For the generated data, the growth of data volume only helps to improve the accuracy of the model when it is small.

experiments in the digits datasets. We use a few target labeled data and the increasing unlabeled data to train the target model. The result is shown in Figure 1(b). For the source data and target data, it is clear that the accuracy of the model trained is higher as the number of the data increases. For the generated data, the growth of data volume only helps to improve the accuracy of the model when it is small (e.g., less than 45 in Figure 1(b)). However, the accuracy of the model fluctuates around 33% regardless of the increase in the unlabeled data, when the number of data exceeds 35. This result shows that the model trained by generate data converge faster than those trained by the source data and target data, since the generated data have less diversity.

In this paper, to show how the diversity of unlabeled data (i.e., the independence among unlabeled data) affects the FHA methods, we theoretically analyze the affect of the sample complexity regarding the FHA problem (Theorem 1). In this analysis, we adopt the log-coefficient score $\alpha$ (Dagan et al., 2019) to measure the dependency among unlabeled data. Our results show that we can still count on the unlabeled data to help address the FHA problem as long as the unlabeled data are weakly dependent ($\alpha < 0.5$). Nevertheless, once $\alpha \geq 0.5$, the results in Theorem 1 may not hold, resulting to fail theoretically. In addition, we find that high dependency among unlabeled data usually means that we need more unlabeled data to obtain a good target-domain classifier. From the above analysis and Figure 1, we argue that diversity matters in addressing the FHA problem.

To this end, we propose the *diversity-enhancing generative network* (DEG-Net) for the FHA problem, which is a weight-shared conditional generative method equipped with a kernel independence measure: the *Hilbert-Schmidt independence criterion* (HSIC) (Gretton et al., 2005; Ma et al., 2020; Pogodin & Latham, 2020), which is used in various situations, e.g., clustering (Song et al., 2007; Blaschko & Gretton, 2008), independence testing (Gretton et al., 2007), and self-supervised classification (Li et al., 2021). Although the log-coefficient score is used to analyze the affect of the sample complexity regarding the FHA problem, its calculation requires to know the distribution regarding the target-domain data, which is unknown in practice. Yet, HSIC can be estimate easily by the data sample. Thus, we adopt the HSIC to calculate the dependency among generated unlabeled data.

The overview of DEG-Net is in Figure 2, showing that there are two modules in DEG-Net: the generation module and the adaptation module. In the generation module, we train the conditional generator with a well-trained source classifier and few target domain data. To train the generator with both the source domain and target domain knowledge and improve the diversity of generated data simultaneously, the generative loss of DEG-Net consists of 3 parts: the classification loss, similarity loss and diversity loss. More specifically, DEG-Net generates data via minimizing the HSIC value (i.e., *maximizing the independence*) between the semantic features of the target data and generated data, where the semantic features are the hidden-layer outputs of the well-trained source hypothesis. To use the generalization knowledge in the semantic features of data that is shared by different classes

(Chen et al., 2020; Chi et al., 2021b; Yao et al., 2021), the generator is a weight-shared network. As for the adaptation module, the source classifier is trained to work well over the target domain. The adaptation module consists of a classifier and a group discriminator (Chi et al., 2021a; Motiian et al., 2017). With the help of the group discriminator which tries to confuse the classifier to distinguish data from different domains, the classifier is trained to classify the data over the target domain using the generated data and few label target data.

We verify the effectiveness of DEG-Net on 8 FHA benchmark tasks (Chi et al., 2021a). Experimental results show that DEG-Net outperforms existing FHA methods and achieves the state-of-the art performance. Besides, due to the weight-shared mechanism, DEG-Net is much faster than previous generative FHA methods in training. We also conduct an ablation study to verify that each component in the DEG-Net is useful, which shows that diverse generated data can help improve the performance when addressing the FHA problem, which lights up a novel road for the FHA problem.

## 2 PRELIMINARY AND RELATED WORKS

**Problem Setting.** In this section, we formalize FHA problem mathematically. Denoting by $\mathcal{X} \subset \mathbb{R}^n$ the input space and by $\mathcal{Y} := \{1, \cdots, K\}$ the output space, where $K$ is the number of classes. The source domain (target domain) can be defined as a joint probability distribution $\mu^s$ on $\mathcal{X} \times \mathcal{Y}$ ($\mu^t$ on $\mathcal{X} \times \mathcal{Y}$). Besides, we assume that there is a well-trained model $f^s : \mathcal{X} \to \mathcal{Y}$. The $f^s$ is trained with data $\{\boldsymbol{x}_i^s, y_i^s\}_{i=1}^n$ drawn *independently and identically distributed* (i.i.d) from $\mu^s$ with the aim of minimizing $\hat{\mathbb{E}}_{(\boldsymbol{x},y) \sim \mu^s}[\ell(h(\boldsymbol{x}), y)]$, where $\ell : \mathcal{Y} \times \mathcal{Y}$ is a loss function to measure if two elements in $\mathcal{Y}$ are close, and $h$ is an element in a hypothesis set $\mathcal{H} := \{h : \mathcal{X} \to \mathcal{Y}\}$. Thus, $f^s$ can be defined as

$$f^s := \arg\min_{h \in \mathcal{H}} \hat{\mathbb{E}}_{(X^s, Y^s)}[\ell(h(X^s), Y^s)]. \tag{1}$$

$f^s$ is also called the *source hypothesis* in our paper. Hence, the FHA problem is defined as follow:

**Problem 1.** *(FHA) Given the source hypothesis $f^s$ defined in Eq. (1) and the labeled dataset $S^t := \{(\boldsymbol{x}_i^t, y_i^t)\}_{i=1}^{m_1}$ ($m_1 \leq 7K$, according to (Chi et al., 2021a; Park et al., 2019)), drawn i.i.d. from target domain $\mu^t$, the FHA methods aim to train a classifier $f^t \in \mathcal{H}$ with $f^s$ and $S^t$ such that we can minimize the value of $\mathbb{E}_{(X^t, Y^t)}[\ell(h(X^t), Y^t)]$, where $h \in \mathcal{H}$. Namely, $f^t = \arg\min_{h \in \mathcal{H}} \mathbb{E}_{(X^t, Y^t)}[\ell(h(X^t), Y^t)]$.*

**Hypothesis Transfer Learning Methods.** *Hypothesis transfer learning* (HTL) aims to train a classifier with only a well-trained classifier and small labeled data or abundant unlabeled data over the target domain (Kuzborskij & Orabona, 2013; Tommasi et al., 2010; Liang et al., 2020). In (Kuzborskij & Orabona, 2013), they used the leave-one-out error to find the optimal transfer parameters based on the regularized least squares with biased regularization. SHOT (Liang et al., 2020) froze the source hypothesis and trained a domian-specific encoding module using the abundant unlabled data. Later, neighborhood reciprocity clustering (Yang et al., 2021a) was proposed to address HTL by encouraging reciprocal neighbors to concord in their label prediction. Different from the FHA problem, the HTL problem does not have the limitation on the number of the target domain labeled data.

**Target-oriented Hypothesis Adaptation Network.** TOHAN (Chi et al., 2021a) is a one-step solution for the FHA problem. It has a good performance due to using the generated unlabeled data in the adaptation process. Motivated by the learnability in *semi-supervised learning* (SSL), TOHAN found that unlabeled data in the intermediate domain, which is compatible with both the source classifier and target classifier, can address the FHA problem for providing the additional information in the training. Guided by this principle, the key module of TOHAN is to generate the unlabeled data drawn from the probability distribution $\mu^m$:

$$\mu^m = \arg\min_{\mu} \left[ \chi(h^s, \mu) + \chi(h^t, \mu) \right] \tag{2}$$

where $\chi(h^s, \mu)$ (resp. $\chi(h^t, \mu)$) measures how compatible $h^s$ (resp. $h^t$) is with unlabeled data distribution $\mu$ (Balcan & Blum, 2010).

## 3 THEORETICAL ANALYSIS REGARDING THE DATA DIVERSITY IN FHA

In previous works, researchers have shown that generated high-compatible data can help address the FHA problem. However, as discussed in Section 1, the diversity of the generated data matters in

addressing the FHA. Besides, the previous studies assume that the generated data is independent in their theory, and is inconsistent with their methods. In this section, we will show how the dependency among the generated data affects the performance of FHA methods. Similar to (Chi et al., 2021a), our theory is also based on the theory regarding SSL (Webb & Sammut, 2010).

**Dependency Measure.** Following Dagan et al. (2019), we use the log-coefficient that measures the dependency among observations from a random variable $Z$ to theoretically analyze the data diversity.

**Definition 1** (Log-influence and log-coefficient (Dagan et al., 2019)). *Let $\boldsymbol{Z} = (Z_1, \ldots, Z_m)$ be a random variable over $(\mathcal{X} \times \mathcal{Y})^m$ and let $\mu_{\boldsymbol{Z}}$ denote either its probability distribution if discrete or its density if continuous. Assume that $\mu_{\boldsymbol{Z}} > 0$ on all $(\mathcal{X} \times \mathcal{Y})^m$. For any $i \neq j \in \{1, 2, \ldots, m\}$, define the log-influence between $j$ and $i$ as*

$$I_{j,i}^{\log}(\boldsymbol{Z}) = \frac{1}{4} \sup_{\substack{Z_{-i-j} \in (\mathcal{X} \times \mathcal{Y})^{m-2} \\ z_i, z_i', z_j, z_j' \in \mathcal{X} \times \mathcal{Y}}} \log \frac{\mu_{\boldsymbol{Z}}[Z_i Z_j Z_{-i-j}] \mu_{\boldsymbol{Z}}[Z_i' Z_j' Z_{-i-j}]}{\mu_{\boldsymbol{Z}}[Z_i' Z_j Z_{-i-j}] \mu_{\boldsymbol{Z}}[Z_i Z_j' Z_{-i-j}]}. \tag{3}$$

*Then the log-coefficient of $\boldsymbol{Z}$ is defined as $\alpha_{\log}(\boldsymbol{Z}) = \max_{i=1,\ldots,m} \sum_{i \neq j} I_{j,i}^{\log}(\boldsymbol{Z})$.*

From Definition 1, it is clear that $\alpha_{\log}(\boldsymbol{Z})$ will be zero if $Z_i$ and $Z_j$ are independent (for any $i \neq j$).

**Sample Complexity Analysis for FHA.** Since the generated data are unlabeled, we follow the theory regarding SSL to analyze how the generated unlabeled data can help address the FHA problem. More importantly, we will analyze how the dependency among the generated data affects the performance of FHA methods. For simplicity, we consider a binary SSL problem (i.e., $K = 2$).

Let $f^* : \mathcal{X} \to \{0, 1\}$ be the optimal target classifier. Let $err(h) = \mathbb{E}_{x \sim \mu_X^t}[h(x) \neq f^*(x)]$ be the true error rate of a hypothesis $h \in \mathcal{H}$ over a marginal distribution $\mu_X^t$. In FHA, its learnability mainly depends on the compatibility $\chi : \mathcal{H} \times \mathcal{X} \mapsto [0, 1]$ measuring how "compatible" $h$ is to one unlabeled data $\boldsymbol{x}$. In the following, we use $\chi(h, \mu_X^t) = \mathbb{E}_{x \sim \mu_X^t}[\chi(h, x)]$ to represent the expectation of compatibility of data from $\mu_X^t$ on a classifier $h$, and let $S_X^{(m_u)}$ be an observation of a random variable $\boldsymbol{X}^{t, m_u} = (X_1^t, \ldots, X_{m_u}^t)$, where the distribution regarding $X_i^t$ is $\mu_X^t$, $i = 1, \ldots, m_u$. The following theorem shows that, under some conditions, we can still learn a good $f^t$ even when the dependency among unlabeled the target data exists.

**Theorem 1.** *Let $\hat{\chi}(h, S_X^{(m_u)}) = \frac{1}{m_u} \sum_{x \in S_X^{(m_u)}} \chi(h, x)$ be the empirical compatibility over $S_X^{(m_u)}$ and $\mathcal{H}_0 = \{h \in \mathcal{H} : \widehat{err}(h) = 0\}$. If $f^* \in \mathcal{H}$, $\chi(f^*, \mu_X^t) = 1 - t$, and $\alpha_{\log}(\boldsymbol{X}^{t, m_u}) < 1/2$, then $m_u$ unlabeled data and $m_l$ labeled data are sufficient to learn to error $\epsilon$ with probability $1 - \delta$, for*

$$m_u = \max\left(\mathcal{O}\left(\frac{1}{(1 - \alpha_{\log}(\boldsymbol{X}^{t, m_u}))\epsilon^2} \log \frac{2}{\delta}\right), \mathcal{O}\left(\frac{\mathrm{VCdim}(\chi(\mathcal{H}))}{(1 - 2\alpha_{\log}(\boldsymbol{X}^{t, m_u}))\epsilon^2}\right)\right) \tag{4}$$

*and*

$$m_l = \frac{2}{\epsilon}\left[\ln(2\mathcal{H}_{\mu_X^t, \chi}(t + 2\epsilon)[2m_l, \mu_X^t]) + \ln\frac{4}{\delta}\right], \tag{5}$$

*where $\chi(\mathcal{H}) = \{\chi_h : h \in \mathcal{H}\}$ is assumed to have a finite VC dimension, $\chi_h(\cdot) = \chi(h, \cdot)$, and $\mathcal{H}_{\mu_X^t, \chi}(t + 2\epsilon)[2m_l, \mu_X^t]$ is the expected number of splits of $2m_l$ data drawn from $\mu_X^t$ using hypotheses in $\mathcal{H}$ of compatibility more than $1 - t - 2\epsilon$. In particular, with probability at least $1 - \delta$, we have $err(\hat{h}) \leq \epsilon$, where $\hat{h} = \arg\max_{h \in \mathcal{H}_0} \hat{\chi}(h, S_X^{(m_u)})$.*

The proof of Theorem 1 is presented in Appendix A, which mainly follows the recent result in the problem of learning from dependent data (Dagan et al., 2019).

Theorem 1 shows that when the dependency among the unlabeled data is weak (i.e., $\alpha_{\log}(\boldsymbol{X}^{t, m_u}) < 1/2$), we can obtain a similar result compared to classical result in the SSL theory (Balcan & Blum, 2010). Namely, if we can generate unlabeled data that are highly compatible to $f^*$, which means that $t$ is very small and thus $\mathcal{H}_{\mu_X^t, \chi}(t + 2\epsilon)[2m_l, \mu_X^t]$ is small, thus we do not need a lot of labeled data to learn a good $f^t$ (Chi et al., 2021a).

**Diversity Matters in FHA.** Theorem 1 also shows that diversity of unlabeled data matters in FHA. There are two reasons. The first reason is that Theorem 1 might not be true if there is strong

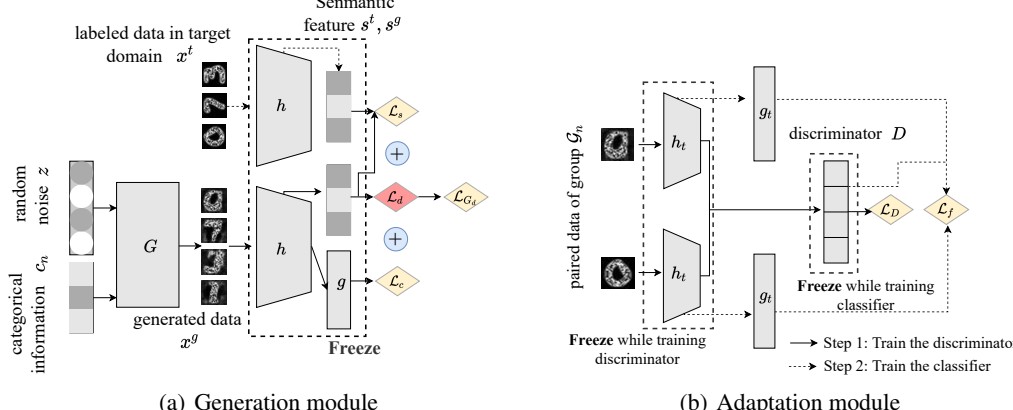

(a) Generation module            (b) Adaptation module

Figure 2: Overview of the *diversity-enhancing generative network* (DEG-Net). It consists of generator $G$, a classifier $f^t = h_t \circ g_t$ (initialize $f^s = f^t$) and group discriminator $D$. (a) In the generation module, we train a generator $G$ with freezing the classifier $f^t = f^s$. (b) In the adaptation module, we first pair the generated data and labeled data and use the paired data to train the discriminator $D$ while freezing the encoder $h_t$. Then, we freeze the discriminator $D$ to train the classifier $f^t$.

dependency among the unlabeled data (e.g., $\alpha_{\log}(\boldsymbol{X}^{t,m_\mathrm{u}}) > 1/2$). This will directly make the previous work lose the theoretical foundation to address the FHA problem. The second reason is that we need more unlabeled data to reach the same error $\epsilon$ if the dependency among the unlabeled data increases. Specifically, if $\alpha_{\log}(\boldsymbol{X}^{t,m_\mathrm{u}})$ is very close to $1/2$, then $m_\mathrm{u}$ could be a very large number. The above reasons motivate us to take care of the dependency among the generated data. To weaken such dependency, we propose our diversity-enhancing generative network for the FHA problem.

## 4    DIVERSITY-ENHANCING GENERATIVE NETWORK FOR FHA PROBLEM

In this section, we propose the *diversity-enhancing generative network* (DEG-Net) for the FHA problem. DEG-Net has two modules: the generation module to generate diverse unlabeled data; the adaptation module to train the classifier for the target domain.

### 4.1    DIVERSITY-ENHANCING GENERATION

In order to overcome the shortcomings of the current generative method for FHA problem, we come up with solutions for both the generative architecture and the loss function. As for the generative architecture, we propose the weight-shared conditional generative network for generating the data with the specific category. As for the generative loss function , we design a novel loss function to constrain the similarities and diversity of the semantic information of the generated data.

**Weight-shared Conditional Generative Network.** As discussed before, for using generalized features which are shared among different classes to improve the quality of generated data and reduce the time of training, the weight-shared conditional generative network is promising. Following Chi et al. (2021a), the generator aims to generate data of specific categories from Gaussian random noise. The encoder outputs the semantic feature and the class probability distribution of the generated data. To achieve the aim of the generative method, we design the two-part loss functions: the classification loss function and the similarity of semantic feature loss function.

We assume that $\boldsymbol{x}_i^{\mathrm{gn}} = G(z^i, c_n)$ is the generated data with a specific category, where the inputs of generator $G$ are the Gaussian random noise $z^i$ and the specific categorical information $c_n$. Specifically, we use the one-hot encoded label as the categorical information. The generated data $\boldsymbol{x}_i^{\mathrm{gn}}$ inputs to the well-trained source-domain classifier $f^s = h_s \circ g_s$, where the output of $h_s$ is the group discriminator feature, which will be used in the adaptation module and the output of $g_s$ is probability feature $\boldsymbol{p_i} = (p_i^1, \ldots p_i^n, \ldots, p_i^K)$, where $p_i^n$ is the probability of the generated data belonging to the $n^{\mathrm{th}}$ class respectively. The semantic feature $s_i^{\mathrm{gn}}$ used in the similarity loss and diversity loss is the hidden-layer output of $h_s$ (details of the hidden-layer selection can be found in Appendix C.). We aim to update parameters of the generator to force the generated data with the categorical information

---

**Algorithm 1** Diversity-enhancing Generative Network (DEG-Net)

---

**Input:** conditional generator $G_{\theta_G}$ parameterized by $\theta_G$, a group discriminator $D_{\theta_D}$ parameterized by $\theta_D$, a classifier $f_{\theta_f}$ parameterized by $\theta_f$, kernel function $k$, generation batch size $B_n$, learning rate $\alpha_G$, $\alpha_D$ and $\alpha_f$, total epoch $T_{max}$, pretraining group discriminator epoch $T_d$.

**for** $t = 1, 2, ....., T_{max}$ **do**

    **for** $n = 0, 1, ..., K-1$ **do**

        **1: Generate** random noise $z$ and categorical information $c_n$;

        **2: Generate** data $G_n(z)$ and then **add** them to $\mathcal{D}_m$;

        **3: Calculate** the semantic feature $s$ and classification probability $p$;

        **4: Calculate** the kernel matrice of semantic feature $S^n$ using kernel function $k$;

    **end**

    **5: Update** $\theta_G \leftarrow \theta_G - \alpha_G \nabla \mathcal{L}_{G_d}(s, p)$ using Eq. (11);

    **if** $t = T_{max} - T_f$ **then**

        **for** $i = 1, 2, ..., T_d$ **do**

            **6: Sample** $\mathcal{G}_1, \mathcal{G}_3$ from $\mathcal{D}_m \times \mathcal{D}_m$;

            **7: Sample** $\mathcal{G}_2, \mathcal{G}_4$ from $\mathcal{D}_m \times \mathcal{D}_t$;

            **8: Update** $\theta_D \leftarrow \theta_D - \alpha_D \nabla \mathcal{L}_D(\{\mathcal{G}_{i=1}^4\})$ using Eq. (12);

        **end**

    **end**

    **if** $t \geq t_{max} - T_f$ **then**

        **9: Sample** $\mathcal{G}_1, \mathcal{G}_3$ from $\mathcal{D}_m \times \mathcal{D}_m$;

        **10: Sample** $\mathcal{G}_2, \mathcal{G}_4$ from $\mathcal{D}_m \times \mathcal{D}_t$;

        **11: Update** $\theta_f \leftarrow \theta_f - \alpha_f \nabla \mathcal{L}_f(\{\mathcal{G}_{i=1}^4\}, x)$ using Eq. (13);

        **12: Update** $\theta_D \leftarrow \theta_D - \alpha_D \nabla \mathcal{L}_D(\{\mathcal{G}_{i=1}^4\})$ using Eq. (12);

    **end**

**end**

**Output:** a well-trained classifier $f_\theta$.

---

$x_i^{\mathrm{gn}}$ belonging to the $n^{\mathrm{th}}$ class, i.e., making $p_n$ close to 1. Specifically, we minimize the following loss to generate the data of a specific category $n$:

$$\mathcal{L}_c = \frac{1}{K} \sum_{n=1}^{K} \frac{1}{B_n} \sum_{i=1}^{B_n} \|p_i^n - 1\|_2^2, \tag{6}$$

where $B_n$ is the batch-size of the generator.

To make the generated data closer to data in the target domain, we need to define the loss function to measure the difference between data of two different domains. Motivated by Zheng & Zhou (2021), DEG-Net uses the semantic features to calculate the similarities. To avoid the copy issue, we decided to use the $\ell_1$ distance $\|\boldsymbol{x} - \boldsymbol{y}\|_1 = \sum_i \omega_i |x_i - y_i|$, where $\omega_i = |x_i - y_i|^2 / \|\boldsymbol{x} - \boldsymbol{y}\|_2$, since it encourages larger gradients for feature dimensions with higher residual errors. Compared to the $\ell_2$-norm, it is better to measure the similarity of the semantic features between the generated images and the target images, since $\ell_1$ distance is more robust to outliers (Oneto et al., 2020). Thus, the similarity loss is defined as following:

$$\mathcal{L}_s = \frac{1}{K} \sum_{n=1}^{K} \frac{1}{m_l M B_n} \sum_{i=1}^{B_n} \sum_{j=1}^{m_l} \left\| \boldsymbol{s}_i^{\mathrm{gn}} - \boldsymbol{s}_j^t \right\|_1, \tag{7}$$

where $M = \max_{\boldsymbol{s}_1, \boldsymbol{s}_2 \in \mathcal{X}} \|\boldsymbol{s}_1 - \boldsymbol{s}_2\|_1$ ($\mathcal{X}$ is compact and $\|\cdot\|_1$ is continuous), $m_l$ is the number of labeled data drawn from the target domain, $\boldsymbol{s}_j^t$ and $\boldsymbol{s}_i^{\mathrm{gn}}$ are the semantic features of target data and generated data, respectively. Combining Eq. (6) and Eq. (7), we obtain the loss to train the weight-shared conditional generative network:

$$\mathcal{L}_G = \mathcal{L}_c + \lambda \mathcal{L}_s, \tag{8}$$

where $\lambda \geq 0$ is a hyper-parameter between two losses. Note that optimizing Eq. (8) is corresponding to TOHAN's principle Eq. (2), where Eq. (6) (resp. Eq. (7)) is corresponding to $\chi(h^s, \mu)$ (resp. $\chi(h^t, \mu)$). To ensure that the conditional generator can generate the image with the correct class label, we pretrain the generator using the well-trained source model for some epochs.

**Generative Function with Diversity.** As discussed above, the weak dependence of unlabeled data is an important condition for using generated unlabeled data to address the FHA problem. To ensure

that the unlabeled data are weakly dependent among unlabeled data (i.e., to generate more diverse unlabeled data), it is necessary to use diversity regularization to train the generator. Unfortunately, the log-coefficient score, a dependence measure used to analyze the sample complexity, is hard to calculate, since its calculation requires the unknown distribution regarding the target-domain data. HSIC, a kernel independence measure can also measure the dependency of the generated data. Different from the log-coefficient score, HSIC can be easily estimated (Gretton et al., 2005; Song et al., 2012):

$$\widehat{\mathrm{HSIC}}(X, Y) = \frac{1}{(N-1)^2} \mathrm{Tr}(KHLH), \tag{9}$$

where $K = (k_{ij}) = k(x_i, x_j)$ $(L = (l_{ij}) = k(y_i, y_j))$ is the kernel matrix ($k(\cdot, \cdot)$ is the kernel function) and $H = I - \frac{1}{N}\mathbf{1}\mathbf{1}^\top$ is the centering matrix. We minimize the HSIC measure of the generated data's semantic features to obtain weakly dependent data. Specifically, we use the Gaussian kernel as the kernel function, and minimize the following loss to generate more diversity data:

$$\mathcal{L}_d = \frac{1}{K} \sum_{n=1}^{K} \sqrt{\widehat{\mathrm{HSIC}}(\boldsymbol{s}^{\mathrm{gn}}, \boldsymbol{s}^{\mathrm{gn}})} = \frac{1}{K} \sum_{n=1}^{K} \sqrt{\frac{1}{(B_n-1)^2} \mathrm{Tr}(S^n H S^n H)}, \tag{10}$$

where $S^n = (\boldsymbol{s}_{ij}^n) = k(\boldsymbol{s}_i^{\mathrm{gn}}, \boldsymbol{s}_j^{\mathrm{gn}})$ is the kernel matrice of the semantic features of the generated data with a specific class. Hence, we obtain the total loss to train the generator with diversity enhancing:

$$\mathcal{L}_{G_d} = \mathcal{L}_G + \beta \mathcal{L}_d, \tag{11}$$

where $\beta \geq 0$ is a hyper-parameter between the generative loss and diversity regularization. Note that the diversity regularization and the similarity loss restrict themselves.

## 4.2 Adaptation Module Using Generated Data

Following Chi et al. (2021a), we create paired data using the labeled data in the target domain and the generated data, and assign the group labels to the paired groups under the following rules: $\mathcal{G}_1$ pairs the generated data with the same class label; $\mathcal{G}_2$ pairs the generated data and the data in the target domain with the same class label; $\mathcal{G}_3$ pairs the generated data but with different class label; $\mathcal{G}_4$ pairs the generated data and the data in the target domain and also with different class label. By using adversarial learning, we train a discriminator $D$ which could distinguish between the data in different domains while maintaining high classification accuracy on generated data. The discriminator $D$ is a four-class classifier with the inputs of the above paired group data. Different from classical adversarial domain adaptation (Ganin et al., 2016; Jiang et al., 2020), the group discriminator $D$ decides which of the four groups a given data pair belongs to. By freezing the encoder, we train $D$ with the cross-entropy loss:

$$\mathcal{L}_D = -\hat{\mathbb{E}} \left[ \sum_{i=1}^{4} y_{\mathcal{G}_i} log(D(\phi(\mathcal{G}_i))) \right], \tag{12}$$

where $\hat{\mathbb{E}}(\cdot)$ represents the empirical mean value, $y_{\mathcal{G}_i}$ is the group label of group $\mathcal{G}_i$ and $\phi(\mathcal{G}_i) := [g(x_1), g(x_2)]$ is the output of the encoder with the paired data input.

Next, we will train the classifier $f^t = h_t \circ g_t$ while freezing the group discriminator, which is initialized with the same weight as that in the source classifier $f^s = h_s \circ g_s$. Motivated by non-saturating games (Goodfellow, 2016), we minimize the following loss to update $f^t$:

$$\mathcal{L}_f = -\gamma \hat{\mathbb{E}} \left[ y_{\mathcal{G}_1} \log \left( D \left( \phi \left( \mathcal{G}_2 \right) \right) \right) - y_{\mathcal{G}_3} \log \left( D \left( \phi \left( \mathcal{G}_4 \right) \right) \right) \right] + \hat{\mathbb{E}} \left[ \ell \left( f_t \left( x_t \right), f_t^* \left( x_t \right) \right) \right], \tag{13}$$

where $\gamma \geq 0$ is a hyper-parameter, $l$ is the cross-entropy loss, and $f_t^*$ is the optimal target model. Note that, the label information of generated data has as certain noise. As demonstrated in Theorem 1, it is only necessary to use generated unlabeled data for addressing the FHA problem. Thus, we only use labeled target data for target supervised loss in Eq. (13).

## 5 Experiments

We compare DEG-Net with previous FHA methods on digits datasets (i.e. MNIST ($M$),USPS ($U$),and SVHN ($S$)) and objects datasets (i.e. CIFAR-10 ($CF$) and STL-10 ($SL$)), following (Chi

Table 1: Classification accuracy±standard deviation (%) on 6 digits FHA tasks. Bold value represents the highest accuracy on each column.

| Tasks | WA | FHA Methods | Number of Target Data per Class | | | | | | |
|---|---|---|---|---|---|---|---|---|---|
| | | | 1 | 2 | 3 | 4 | 5 | 6 | 7 |
| $M \to U$ | 69.7 | FT | 70.2±0.0 | 70.6±0.3 | 70.7±0.1 | 70.8±0.3 | 70.9±0.2 | 71.1±0.3 | 71.1±0.4 |
| | | SHOT | 72.6±1.9 | 73.6±2.0 | 74.1±0.6 | 74.6±1.2 | 74.9±0.7 | 75.4±0.3 | 76.1±1.5 |
| | | S+FADA | 74.4±1.5 | 83.1±0.7 | 83.3±1.1 | 85.9±0.5 | 86.0±1.2 | 87.6±2.6 | 89.1±1.0 |
| | | T+FADA | 74.2±1.8 | 81.6±4.0 | 83.4±0.8 | 82.0±2.3 | 86.2±0.7 | 87.2±0.8 | 88.2±0.6 |
| | | TOHAN | 76.0±1.9 | 83.3±0.3 | 84.2±0.4 | 86.5±1.1 | 87.1±1.3 | 88.0±0.5 | 89.7±0.5 |
| | | DEG-Net | **83.1±0.9** | **86.2±0.8** | **86.5±0.6** | **88.7±0.9** | **89.6±0.5** | **91.5±0.6** | **92.1±0.6** |
| $M \to S$ | 24.1 | FT | 26.7±1.0 | 26.8±2.1 | 26.8±1.6 | 27.0±0.7 | 27.3±1.2 | 27.5±0.8 | 28.3±1.5 |
| | | SHOT | 25.7±2.2 | 26.9±1.2 | 27.9±2.6 | 29.1±0.4 | 29.1±1.4 | 29.6±1.7 | 29.8±1.5 |
| | | S+FADA | 25.6±1.3 | 27.7±0.5 | 27.8±0.7 | 28.2±1.3 | 28.4±1.4 | 29.0±1.0 | 29.6±1.9 |
| | | T+FADA | 25.3±1.0 | 26.3±0.8 | 28.9±1.0 | 29.1±1.3 | 29.2±1.3 | 31.9±0.4 | 32.4±1.8 |
| | | TOHAN | 26.7±0.1 | **28.6±1.1** | 29.5±1.4 | 29.6±0.4 | 30.5±1.2 | 32.1±0.2 | 33.2±0.8 |
| | | DEG-Net | **27.2±0.3** | 28.5±1.3 | **29.7±0.9** | **30.7±0.8** | **32.9±1.5** | **33.7±1.8** | **34.9±1.6** |
| $S \to U$ | 64.3 | FT | 64.9±1.1 | 66.5±1.5 | 66.7±1.7 | 67.3±1.1 | 68.1±2.3 | 68.3±0.5 | 69.7±1.4 |
| | | SHOT | 74.7±0.3 | 75.5±1.4 | 75.6±1.0 | 75.8±0.7 | 77.1±2.1 | 77.8±1.6 | 79.6±0.6 |
| | | S+FADA | 72.2±1.4 | 73.6±1.4 | 74.7±1.4 | 76.2±1.3 | 77.2±1.7 | 77.8±3.0 | 79.7±1.9 |
| | | T+FADA | 71.7±0.6 | 74.3±1.9 | 74.5±0.8 | 75.9±2.1 | 77.7±1.5 | 76.8±1.8 | 79.7±1.9 |
| | | TOHAN | **75.8±0.9** | **76.8±1.2** | **79.4±0.9** | 80.2±0.6 | 80.5±1.4 | 81.1±1.1 | 82.6±1.9 |
| | | DEG-Net | 75.2±0.3 | 76.9±1.5 | 78.2±1.2 | **80.7±1.5** | **81.7±1.7** | **83.1±1.7** | **84.3±2.2** |
| $S \to M$ | 70.2 | FT | 70.2±0.0 | 70.6±0.3 | 70.7±0.1 | 70.8±0.3 | 70.9±0.2 | 71.1±0.3 | 71.1±0.4 |
| | | SHOT | 72.6±1.9 | 73.6±2.0 | 74.1±0.6 | 74.6±1.2 | 74.9±0.7 | 75.4±0.3 | 76.1±1.5 |
| | | S+FADA | 74.4±1.5 | 83.1±0.7 | 83.3±1.1 | 85.9±0.5 | 86.0±1.2 | 87.6±2.6 | 89.1±1.0 |
| | | T+FADA | 74.2±1.8 | 81.6±4.0 | 83.4±0.8 | 82.0±2.3 | 86.2±0.7 | 87.2±0.8 | 88.2±0.6 |
| | | TOHAN | **76.0±1.9** | **83.3±0.3** | 84.2±0.4 | **86.5±1.1** | 87.1±1.3 | 88.0±0.5 | 89.7±0.5 |
| | | DEG-Net | 76.2±1.3 | 78.2±1.3 | **85.7±0.6** | 85.9±0.8 | **88.6±1.6** | **89.5±1.2** | **90.2±0.7** |
| $U \to M$ | 82.9 | FT | 83.5±0.4 | 84.3±2.4 | 84.5±0.7 | 85.5±1.3 | 86.6±1.0 | 87.2±0.7 | 88.1±2.7 |
| | | SHOT | 83.1±0.5 | 85.5±0.3 | 85.8±0.6 | 86.0±0.2 | 86.6±0.2 | 86.7±0.2 | 87.0±0.1 |
| | | S+FADA | 83.2±0.2 | 84.0±0.3 | 85.0±1.2 | 85.6±0.5 | 85.7±0.6 | 86.2±0.6 | 87.2±1.1 |
| | | T+FADA | 82.9±0.7 | 83.9±0.2 | 84.7±0.8 | 85.4±0.6 | 85.6±0.7 | 86.3±0.9 | 86.6±0.7 |
| | | TOHAN | **84.0±0.5** | 85.2±0.3 | 85.6±0.7 | 86.5±0.5 | 87.3±0.6 | 88.2±0.7 | 89.2±0.5 |
| | | DEG-Net | 82.2±0.7 | **85.9±0.6** | **86.5±1.5** | **87.8±0.9** | **88.9±0.9** | **90.3±0.5** | **91.6±1.2** |
| $U \to S$ | 17.3 | FT | 23.4±1.8 | 23.6±2.7 | 23.8±1.6 | 24.6±1.4 | 24.6±1.2 | 24.8±0.7 | 25.5±1.8 |
| | | SHOT | **30.3±1.2** | **31.6±0.4** | 29.8±0.5 | 29.4±0.3 | 29.7±0.5 | 29.8±0.8 | 30.1±0.9 |
| | | S+FADA | 28.1±1.2 | 28.7±1.3 | 29.0±1.2 | 30.1±1.1 | 30.3±1.3 | 30.7±1.0 | 30.9±1.5 |
| | | T+FADA | 27.5±1.4 | 27.9±0.9 | 28.4±1.3 | 29.4±1.8 | 29.5±0.7 | 30.2±1.0 | 30.4±1.7 |
| | | TOHAN | 29.9±1.2 | 30.5±1.2 | 31.4±1.1 | 32.8±0.9 | 33.1±1.0 | 34.0±1.0 | 35.1±1.8 |
| | | DEG-Net | 29.1±1.3 | 30.7±1.1 | **31.8±0.7** | **33.0±1.6** | **33.5±1.4** | **35.1±1.3** | **36.2±1.2** |

et al., 2021a). Following the standard domain-adaptation protocols (Shu et al., 2018), we compare DEG-Net with 4 baselines: (1) *Without adaptation* (WA); (2) *Fine tuning* (FT); (3) *SHOT* (Liang et al., 2020); (4) *S+FADA* (Chi et al., 2021a); (5) *T+FADA* (Chi et al., 2021a); and (6) *TOHAN* (Chi et al., 2021a). Details regarding datasets, baselines and implementation are in Appendixes B and C.

**Digits Datasets.** Following Chi et al. (2021a); Motiian et al. (2017), We conduct 6 tasks of the adaptation among the 3 digital datasets and choose the number of target data from 1 to 7 per class. The classifier accuracy on the target domain of our method over 6 tasks is shown in Table 1. The results show that the performance of DEG-Net is the best on almost all the tasks. It is clear that the accuracy of DEG-Net is lower than TOHAN when the amount of target data is too small. The diversity regularization and the similarity loss restrict each other, to avoid the copy issue. However, when the amount of target data is too small, the target domain information is few, so the generator is less likely to generate similar data with the target domain. Diversity loss enhances this adversarial effect, resulting DEG-Net degrading to TOHAN and SHOT. Another improvement of DEG-Net over TOHAN is that the faster training process of the generator. We need 0.93s to complete the training within each epoch in DEG-Net, while needing 1.35s in TOHAN.

**Objects Datasets.** Following Chi et al. (2021a), we examine the performance of DEG-Net on 2 object tasks and choose the number of target data as 10 per class. The classification accuracies on object tasks are shown in Table 2. It is clear that we outperform baselines. In $CF \to SL$, we achieve 1.5% improvement over TOHAN. In $SL \to CF$, we achieve a performance accracy of 57.2%, 0.3% improvement over S+FADA. It is clear that the effect of DEG-Net is not obvious in objective tasks. It may be caused by the simple structure of generative networks and complexity of datasets.

Table 2: Classification accuracy±standard deviation (%) on 2 objects FHA tasks. Bold value represents the highest accuracy on each row.

| Tasks | FHA Methods | | | | | | |
| --- | --- | --- | --- | --- | --- | --- | --- |
| | WA | FT | SHOT | S+FADA | T+FADA | TOHAN | DEG-Net |
| $CF \to SL$ | 70.6 | 71.5±1.0 | 71.9±0.4 | 72.1±0.4 | 71.3±0.5 | 72.8±0.1 | **74.3±0.3** |
| $SL \to CF$ | 51.8 | 54.3±0.5 | 53.9±0.2 | 56.9±0.5 | 55.8±0.8 | 56.6±0.3 | **57.2±0.5** |

Table 3: Ablation study. Classification accuracy±standard deviation(%) on $M \to U$. Bold value represents the highest accuracy on each column.

| FHA Methods | Number of Target Data per Class | | | | | | |
| --- | --- | --- | --- | --- | --- | --- | --- |
| | 1 | 2 | 3 | 4 | 5 | 6 | 7 |
| TOHAN | 76.0±1.9 | 83.3±0.3 | 84.2±0.4 | 86.5±1.1 | 87.1±1.3 | 88.0±0.5 | 89.7±0.5 |
| separate generative DEG-Net | 75.7±0.7 | 84.7±0.5 | 85.0±1.2 | 85.9±0.9 | 87.4±0.8 | 89.1±1.0 | 90.4±1.2 |
| DEG-Net w/o diversity | 87.2±1.9 | **89.5±0.3** | 89.2±0.4 | 90.2±1.1 | 90.3±1.3 | 91.1±0.5 | 91.2±0.5 |
| DEG-Net | **87.3±0.9** | 89.2±0.8 | **90.1±0.6** | **90.8±0.9** | **90.6±0.5** | **91.5±0.6** | **92.1±0.6** |

**DEG-Net Generates More Diverse Data Than TOHAN.** In this part, we analyze the diversity of the generated data by DEG-Net and TOHAN to see if our generation process can produce more diverse data than TOHAN's. We choose the square root of the HSIC to measure the diversity of the generated data in the task $M \to S$, and calculate the HSIC value among the target-domain data as a reference value that is $0.0013$. After the calculation, the average diversity measure of DEG-Net is $0.0019$, and the average diversity measure of TOHAN is $0.0027$. It is clear that DEG-Net can generate more diverse data than TOHAN. The detailed diversity analysis can be found in Appendix D.

**Ablation Study.** To show the advantage of weight-shared architecture and the diversity loss, we conduct two experiments: (1) The architecture of weight-shared is the same as the DEG-Net but uses Eq. (8) to train the generator (DEG-Net without diversity). (2) The separate generative method, which is similar to TOHAN, has $K$ generators and use the semantic features to calculate the similarity loss for training each generator:

$$\mathcal{L}_{G_s}^n = \frac{1}{B_n} \|p_n - 1\|_2^2 + \lambda \frac{1}{N_y M B_n} \sum_{i=1}^{B_n} \sum_{j=1}^{N_y} \left\| s_i^{gn} - s_j^t \right\|_1. \tag{14}$$

As shown in Table 3, DEG-Net works better than both methods introduced above, and the weight-shared architecture works better than the separate generative method, which reveals that both the weight-shared architecture and the diversity loss can improve the quality of generated data and thus achieve the higher accuracy. Specifically, compared to modified DEG-Net without the diversity loss, the separate generative method ignores the generalization knowledge in the semantic features of data which is shared with all the classes. Modified DEG-Net discards the diversity loss, and thus generates the low diverse data and results in the worse performance. However, the HSIC diversity loss does not work for all the situations. It is clear that DEG-Net achieves the similar accuracy with mo dified DEG-Net without diversity or even worse if the amount of the labeled data is very few (i.e., $m_1 \leq 2$). This phenomenon may be caused by worse data generated by diversity method. Since the diversity loss restricting to the similarity loss, the generator is less likely to generated similar data over target domain (i.e., the distribution of generated data is far from the target domain).

## 6 CONCLUSION

In this paper, we focus on generating more diverse unlabeled data for addressing the FHA problem. We experimentally and theoretically prove that the diversity of generated data (i.e., the independence among the generated data) matters in addressing the FHA problem. For addressing FHA problem, we propose a *diversity-enhancing generative network* (DEG-Net), which consists of the generation module and the adaptation module. With the weight-shared conditional generative method equipped with a kernel independence measure: HSIC, DEG-Net can generate more diverse unlabeled data and achieve the better performance. Experiments show that the generated data of DEG-Net are more diverse, and thus DEG-Net achieves the state-of-the art performance when addressing the FHA problem, which lights up a novel and theoretical-guaranteed road to the FHA problem in the future.

## 7 REPRODUCIBILITY STATEMENT

We implement all methods by PyTorch 1.7.1 and Python 3.7.6, and conduct all the experiments on two NVIDIA RTX 2080Ti GPUs. We use the standard DCGAN network (Radford et al., 2015) as the generator network architecture. We adopt the backbone network of LeNet-5 (LeCun et al., 1998) with batch normalization and dropout as the encoder. We employ connected layers with softmax function as the classifier. The semantic feature in digital tasks is the output of first fully connection layer. We adopt 3 connected layers with softmax function as the group discriminator $D$. We choose the Gaussian kernel as the kernel funcion to calculate the HSIC measure. The hyper-parameter settings details can be found in Appendix C. The details regarding to the datasets used in the paper can be found in Appendix B. For the theoretical resultsclear explanations, we proof the Theorem 1 in Appendix A.

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

# A PROOF OF THEOREM 1

Before proving the Theorem 1, we first introduce a McDiarmid-like inequality under the log-coefficient of a random vector $\boldsymbol{Z}$.

**Lemma 1** (McDiarmid-like Inequality under the Log-coefficient of $\boldsymbol{Z}$). *Let $\mu^{(m)}$ be a distribution defined over $\mathcal{Z}^m$, and $\boldsymbol{Z} = (Z_1, \ldots, Z_m) \sim \mu^{(m)}$ be a random vector, and $g : \mathcal{Z}^m \to \mathbb{R}$ with the following bounded differences property with parameters $\lambda_1, \ldots, \lambda_m > 0$:*

$$\forall Z_i, Z_j : \ |g(Z_i) - g(Z_j)| \leq \sum_{i=1}^{m} 1_{Z_i \neq Z_j} \lambda_i, \tag{15}$$

*where $\mathcal{Z}^m = (\mathcal{X} \times \mathcal{Y})^m$. If $\alpha_{\log}(\boldsymbol{Z}) < 1$, then, for all $t > 0$,*

$$\Pr[|g(\boldsymbol{Z}) - \mathbb{E}[g(\boldsymbol{Z})]| \geq t] \leq 2 \exp\left(-\frac{(1 - \alpha_{\log}(\boldsymbol{Z}))t^2}{2\sum_{i=1}^{m} \lambda_i^2}\right). \tag{16}$$

*Proof.* Based on Definition 2.2 and Lemma 5.2 in (Dagan et al., 2019), we know that $\mu^{(m)}$ satisfies Dobrushin's condition with a coefficient $\alpha < 1$. Thus, based on Theorem 2.3 in (Dagan et al., 2019), we have

$$\Pr[|g(\boldsymbol{Z}) - \mathbb{E}[g(\boldsymbol{Z})]| \geq t] \leq 2 \exp\left(-\frac{(1 - \alpha)t^2}{2\sum_{i=1}^{m} \lambda_i^2}\right). \tag{17}$$

Since, $\alpha \leq \alpha_{\log}(\boldsymbol{Z})$, we prove this lemma. $\qquad\square$

Then, we introduce a recent result regarding bounding the expected suprema of a empirical process using the corresponding Gaussian complexity.

**Theorem 2** ((Dagan et al., 2019)). *Let $\boldsymbol{Z}$ be a random vector over some domain $\mathcal{Z}^m$ and let $\mathcal{G}$ be a class of functions from $\mathcal{Z}$ to $\mathbb{R}$. If $\alpha_{\log}(\boldsymbol{Z}) < 1/2$, then*

$$\mathbb{E}_{S \sim \boldsymbol{Z}} \sup_{g \in \mathcal{G}} \left(\frac{1}{m}\sum_{i=1}^{m} g(s_i) - \mathbb{E}_{S}\left[\frac{1}{m}\sum_{i=1}^{m} g(s_i)\right]\right) \leq \frac{C\mathfrak{G}_{\boldsymbol{Z}}(\mathcal{G})}{\sqrt{1 - 2\alpha_{\log}(\boldsymbol{Z})}}, \tag{18}$$

*where $C > 0$ is a universal constant, and $S = (s_1, \ldots, s_m)$ is a sample of $\boldsymbol{Z}$.*

Note that, the above result is very general, it does not assume that the $m$ marginals of the distribution of $\boldsymbol{Z}$ are identical. Based on the above theorem and lemma, we can prove the following lemma.

**Lemma 2.** *Let $\boldsymbol{Z}$ be a random vector over some domain $\mathcal{Z}^m$ and let $\mathcal{G}$ be a class of functions from $\mathcal{Z}$ to $\mathbb{R}$. If $\alpha_{\log}(\boldsymbol{Z}) < 1/2$, and there exists $L > 0$ such that for any $g \in \mathcal{G}$ and $Z_i$, $|g(Z_i)| \leq L$, then, for any $t > 0$,*

$$\Pr_{S \sim \boldsymbol{Z}}\left[\sup_{g \in \mathcal{G}} \left|\frac{1}{m}\sum_{i=1}^{m} g(s_i) - \mathbb{E}_{S}\left[\frac{1}{m}\sum_{i=1}^{m} g(s_i)\right]\right| \geq \frac{C\mathfrak{G}_{\boldsymbol{Z}}(\mathcal{G})}{\sqrt{1 - 2\alpha_{\log}(\boldsymbol{Z})}} + t\right] \leq e^{-\frac{(1 - \alpha_{\log}(\boldsymbol{Z}))mt^2}{C'L^2}} \tag{19}$$

*for some universal constants $C, C' > 0$.*

*Proof.* We first consider proving the non-absolute-value version. Let

$$M(S) = \sup_{g \in \mathcal{G}} \left(\frac{1}{m}\sum_{i=1}^{m} g(s_i) - \mathbb{E}_{S}\left[\frac{1}{m}\sum_{i=1}^{m} g(s_i)\right]\right). \tag{20}$$

For any $S \sim \boldsymbol{Z}$ and $S' \sim \boldsymbol{Z}$, we have that $|M(S) - M(S')| \leq \sum_{i=1}^{m} 2L 1_{s_i \neq s'_i}/m$. According to Lemma 1, we have

$$\Pr_{S \sim \boldsymbol{Z}}[M(S) - \mathbb{E}[M(S)] \geq t] \leq \exp\left(-\frac{(1 - \alpha_{\log}(\boldsymbol{Z}))mt^2}{C'L^2}\right) \tag{21}$$

for some universal constant $C' > 0$. Then, combining $\mathbb{E}[M(S)]$ (based on Theorem 2), we have

$$\Pr_{S \sim \mathbf{Z}}\left[\sup_{g \in \mathcal{G}}\left(\frac{1}{m}\sum_{i=1}^{m}g(s_i) - \mathbb{E}_S\left[\frac{1}{m}\sum_{i=1}^{m}g(s_i)\right]\right) \geq \frac{C\mathfrak{G}_{\mathbf{Z}}(\mathcal{G})}{\sqrt{1 - 2\alpha_{\log}(\mathbf{Z})}} + t\right] \leq e^{-\frac{(1-\alpha_{\log}(\mathbf{Z}))mt^2}{C'L^2}}. \tag{22}$$

For the opposite inequality, $-M(S)$ part, following (Dagan et al., 2019), we can apply the same arguments on $-\mathcal{G}$. Note that $\mathfrak{G}(-\mathcal{G}) = \mathfrak{G}(\mathcal{G})$, which concludes the bound. $\qquad\square$

The above lemma is a slightly general version of Theorem 6.7 in (Dagan et al., 2019) by considering the influence of $\alpha_{\log}(\mathbf{Z})$. Based on Lemma 2, we can prove the Theorem 1 below.

**Theorem 1.** *Let* $\hat{\chi}(h, S_X^{(m_\mathrm{u})}) = \frac{1}{m_\mathrm{u}}\sum_{x \in S_X^{(m_\mathrm{u})}}\chi(h, x)$ *be the empirical compatibility over* $S_X^{(m_\mathrm{u})}$ *and* $\mathcal{H}_0 = \{h \in \mathcal{H} : \widehat{err}(h) = 0\}$. *If* $f^* \in \mathcal{H}$, $\chi(f^*, \mu_X^t) = 1 - t$, *and* $\alpha_{\log}(\mathbf{X}^{t,m_\mathrm{u}}) < 1/2$, *then* $m_u$ *unlabeled data and* $m_l$ *labeled data are sufficient to learn to error* $\epsilon$ *with probability* $1 - \delta$, *for*

$$m_\mathrm{u} = \max\left(\mathcal{O}\left(\frac{1}{(1 - \alpha_{\log}(\mathbf{X}^{t,m_\mathrm{u}}))\epsilon^2}\log\frac{2}{\delta}\right), \mathcal{O}\left(\frac{\mathrm{VCdim}(\chi(\mathcal{H}))}{(1 - 2\alpha_{\log}(\mathbf{X}^{t,m_\mathrm{u}}))\epsilon^2}\right)\right) \tag{4}$$

*and*

$$m_\mathrm{l} = \frac{2}{\epsilon}\left[\ln(2\mathcal{H}_{\mu_X^t,\chi}(t + 2\epsilon)[2m_\mathrm{l}, \mu_X^t]) + \ln\frac{4}{\delta}\right], \tag{5}$$

*where* $\chi(\mathcal{H}) = \{\chi_h : h \in \mathcal{H}\}$ *is assumed to have a finite VC dimension,* $\chi_h(\cdot) = \chi(h, \cdot)$, *and* $\mathcal{H}_{\mu_X^t,\chi}(t+2\epsilon)[2m_\mathrm{l}, \mu_X^t]$ *is the expected number of splits of* $2m_\mathrm{l}$ *data drawn from* $\mu_X^t$ *using hypotheses in* $\mathcal{H}$ *of compatibility more than* $1 - t - 2\epsilon$. *In particular, with probability at least* $1 - \delta$, *we have* $err(\hat{h}) \leq \epsilon$, *where* $\hat{h} = \arg\max_{h \in \mathcal{H}_0}\hat{\chi}(h, S_X^{(m_\mathrm{u})})$.

*Proof.* Let $S$ be the set of $m_\mathrm{u}$ unlabeled data. Based on the relation between VC dimension and the Gaussian complexity, Lemma 2 gives that, with probability at least $1 - \frac{\delta}{2}$, we have

$$|\Pr_{x \sim \bar{S}}[\chi_h(x) = 1] - \Pr_{x \sim \mu_X^t}[\chi_h(x) = 1]| \leq \epsilon \quad \text{for all } \chi_h \in \chi(\mathcal{H}),$$

where $\bar{S}$ denotes the uniform distribution over $S$. Since $\chi_h(x) = \chi(h, x)$, this implies that we have

$$|\chi(h, D) - \hat{\chi}(h, S)| \leq \epsilon \quad \text{for all } h \in \mathcal{H}.$$

Therefore, the set of hypotheses with $\hat{\chi}(h, S) \geq 1 - t - \epsilon$ is contained in $\mathcal{H}_{\mu_X^t,\chi}(t + 2\epsilon)$.

The bound on the number of labeled data now follows directly from known concentration results using the expected number of partitions instead of the maximum in the standard VC-dimension bounds. This bound ensures that with probability $1 - \frac{\delta}{2}$, none of the functions $h \in \mathcal{H}_{\mu_X^t,\chi}(t + 2\epsilon)$ with $err(h) \geq \epsilon$ have $\widehat{err}(h) = 0$.

The above two arguments together imply that with probability $1 - \delta$, all $h \in \mathcal{H}$ with $\widehat{err}(h) = 0$ and $\hat{\chi}(h, S) \geq 1 - t - \epsilon$ have $err(h) \geq \epsilon$, and furthermore $f^*$ has $\hat{\chi}(f^*, S) \geq 1 - t - \epsilon$. This in turn implies that with probability at least $1 - \delta$, we have $err(\hat{h}) \leq \epsilon$, where $\hat{h} = \arg\max_{h \in \mathcal{H}_0}\hat{\chi}(h, S)$. $\qquad\square$

## B   DATASETS

**Digits.**   Following TOHAN(Chi et al., 2021a), we conduct 6 adaptation experiments on digits datasets: $M \to U$, $M \to S$, $S \to U$, $S \to M$, $U \to M$ and $U \to S$. MNIST ($M$) (LeCun et al., 1998) is the handwritten digits dataset, which have been size-normalized adn centered in $28 \times 28$ pixels. SVHN ($S$) (Netzer et al., 2011) is the real-world image digits dataset, of which images are $32 \times 32$ pixels with 3 channels. USPS ($U$) (Hull, 1994) data are 16×16 grayscale pixels. The SVHN and USPS images are resized to $28 \times 28$ grayscale pixels in the adaptation task (Chi et al., 2021a).

**Objects.**   Following (Sun et al., 2019), we compared DEG-Net and benchmark on CIFAR-10 and STL-10. The CIFAR-10 (Krizhevsky et al., 2009) dataset contains $60,000$ $32 \times 32$ color images in 10 categories, while the STL-10 (Coates et al., 2011) dataset is inspired by the CIFAR-10 dataset with some modifications. However, these two datasets only contain nine overlapping classes. We removed the non-overlapping classes ("frog" and "monkey") (Shu et al., 2018).

## C   DETAILS REGARDING EXPERIMENTS

**Baselines.** We follow the standard domain-adaptation protocols (Shu et al., 2018) and compare DEG-Net with 4 baseline: (1) *Without adaptation* (WA): to classify target domain with the well-trained source domain calssifier. (2) *Fine tuning* (FT): to train the last connected layer of the classifier with few accessible labeled data. (3) *SHOT*: an HTL mehtod, where we modify it to use both the labeled target data and unlabeled target data (Liang et al., 2020). (4) *S+FADA*:to generate unlabeled data using the loss $\mathcal{L}_c$ with the well-trained source clasifier and apply them into DANN (Ganin et al., 2016). (5) *T+FADA*:to generate unlabeled data using the loss $\mathcal{L}_s$ with the few labeled target data and apply them into DANN. (6) *TOHAN*: a novel FHA method, which generate the specific category unlabeled data separately (Chi et al., 2021a).

**Implementation Details.** We implement all methods by PyTorch 1.7.1 and Python 3.7.6, and conduct all the experiments on NVIDIA RTX 2080Ti GPUs. Due to the limitation of the accessible computing resources, we can not choose more complex networks as the backbone of the generator.

Our conditional generator $G$ uses the standard DCGAN network (Radford et al., 2015). We adopt the backbone network of LeNet-5 with batch normalization and dropout to extarct the group discriminator feature. We employ connected layers with softmax function as the classifier to obtain the probability. The semantic feature in digital tasks is the output of first fully connection layer. We adopt 3 connected layers with softmax function as the group discriminator $D$.

**Hyper-parameter Settings.** Following the common protocol of domain adaptation (Shu et al., 2018), we set fixed hyper-parameters for the different datasets. We pretrain the conditional generator for 300 epochs and pretrain the group discriminator for 100 epochs. The training step of the classifier (i.e. the adaptation module) are set 50. As for the generator and the group discriminator, the learding rate of adam optimizer is set to $1 \times 10^{-3}$. As for the classifier, the learding rate of adam optimizer is set to $1 \times 10^{-2}$. The tradeoff parameter $\lambda$ in Eq. (8) is set to 0.9 and the tradeoff parameter $\beta$ in Eq. (11) is set to 0.1. Following (Long et al., 2018) the radeoff parameter $\gamma$ in Eq. (13) is set to $\frac{2}{1+\exp(-10\dot{q})} - 1$.

## D   ADDITIONAL ANALYSIS

**Augmentation Techniques on the FHA Problem**   In this section, we compare the accuracy of the target classifier trained by TOHAN and that of TOHAN with the basic geometric data augmentation for FHA problem over the digit tasks. The geometric data augmentation technique has been widely-explored to diversify the image data (Shorten & Khoshgoftaar, 2019). In our experiment, we randomly choose one or more the following augmentation technique: resizing, shifting, cropping and slight rotations (1 and 20 and -1 to -20) fot the generated data in TOHAN. The classifier accuracy on the target domain of our method over 4 experiments and the average accuracy is shown in Table 4.

It is clear that the performance of the augmentation techniques is worse than our method in general. It may be caused by the fact that the generated image are similar and even the same as the few target data. The diversity of is still low with data augmentation. The accuracy of the augmentation is basically the same as TOHAN's. The improvement brought by the augmentation is more obvious while the number of the target data is increasing.

**Diversity Analysis of DEG-Net**   In this section, we compare the diversity of generated data of DEG-Net with that of TOHAN and target data. Because of the difficulty of calculating log-influence, we use the HSIC to measure the diversity of data. Considering that the generated batch in the training process is 32, we calculate the HSIC measure with the 32 sample data. Table 5 shows the diversity of the different data. It is clear that the diversity loss in DEG-Net works well to make the generated data data more diverse.

**Data Efficiency Analysis of DEG-Net**   In this section, we conduct the experiments in the taskS $M \to S$ and $M \to U$ to analyze the efficiency of the generated data. Following the architecture of the DEG-Net, we use the Eq. (11) to train the conditional generator and obtain the following loss to

Table 4: Classification accuracy±standard deviation (%) on digits FHA tasks of the data augmentation. Bold value represents the highest average accuracy on each column.

| Method | Tasks | Number of Target Data per Class | | | | | | |
|---|---|---|---|---|---|---|---|---|
| | | 1 | 2 | 3 | 4 | 5 | 6 | 7 |
| TOHAN with augmentation | $M \to U$ | 77.1±0.4 | 83.5±0.6 | 84.0±0.7 | 86.7±1.1 | 87.5±0.6 | 88.1±1.4 | 89.4±1.1 |
| | $M \to S$ | 26.7±1.0 | 27.8±1.6 | 29.7±1.3 | 29.4±0.7 | 30.3±1.2 | 32.4±0.8 | 33.5±1.5 |
| | $S \to M$ | 76.4±0.5 | 78.6±0.3 | 82.7±0.1 | 86.5±0.3 | 87.9±0.2 | 88.2±0.3 | 89.6±0.4 |
| | $U \to M$ | 82.1±0.7 | 84.9±1.3 | 85.3±0.6 | 86.7±1.5 | 87.4±0.8 | 87.9±0.7 | 89.8±0.4 |
| | Average of 4 tasks | 65.6±0.7 | 68.7±1.0 | 70.4±0.7 | 72.4±0.9 | 73.3±0.7 | 74.1±0.8 | 75.6±0.8 |
| TOHAN | $M \to U$ | 76.0±1.9 | 83.3±0.3 | 84.2±0.4 | 86.5±1.1 | 87.1±1.3 | 88.0±0.5 | 89.7±0.5 |
| | $M \to S$ | 26.7±0.1 | 28.6±1.1 | 29.5±1.4 | 29.6±0.4 | 30.5±1.2 | 32.1±0.2 | 33.2±0.8 |
| | $S \to M$ | 76.0±1.9 | 83.3±0.3 | 84.2±0.4 | 86.5±1.1 | 87.1±1.3 | 88.0±0.5 | 89.7±0.5 |
| | $U \to M$ | 84.0±0.5 | 85.2±0.3 | 85.6±0.7 | 86.5±0.5 | 87.3±0.6 | 88.2±0.7 | 89.2±0.5 |
| | Average of 4 tasks | 65.7±1.1 | **70.1±0.5** | 70.9±0.7 | 72.2±0.8 | 73.0±1.1 | 74.0±0.5 | 75.5±0.6 |
| DEG-Net | $M \to U$ | 83.1±0.9 | 86.2±0.8 | 86.5±0.6 | 88.7±0.9 | 89.6±0.5 | 91.5±0.6 | 92.1±0.6 |
| | $M \to S$ | 27.2±0.3 | 28.5±1.3 | 29.7±0.9 | 30.7±0.8 | 32.9±1.5 | 33.7±1.8 | 34.9±1.6 |
| | $S \to M$ | 76.2±1.3 | 78.2±1.3 | 85.7±0.6 | 85.9±0.8 | 88.6±1.6 | 89.5±1.2 | 90.2±0.7 |
| | $U \to M$ | 82.2±0.7 | 85.9±0.6 | 86.5±1.5 | 87.8±0.9 | 88.9±0.9 | 90.3±0.5 | 91.6±1.2 |
| | Average of 4 tasks | **67.1±0.8** | 69.7±1.0 | **72.1±0.9** | **73.3±0.9** | **75.0±1.1** | **76.3±1.0** | **77.2±1.0** |

Table 5: The Diversity of the target data and generated data by different methods.

| Task | Target Data | TOHAN | DEG-Net |
|---|---|---|---|
| $M \to S$ | | 0.0027 | **0.0019** |
| $U \to S$ | 0.0013 | 0.0025 | **0.0021** |
| $S \to M$ | | 0.0016 | **0.0013** |
| $U \to M$ | 0.0004 | 0.0014 | **0.0008** |
| $S \to U$ | | 0.0012 | **0.001** |
| $M \to U$ | 0.0002 | 0.0009 | **0.0005** |

update classifier $f_t$:

$$
\begin{aligned}
\mathcal{L}^*{}_f = & -\gamma \hat{\mathbb{E}} \left[ y_{\mathcal{G}_1} \log \left( D \left( \phi \left( \mathcal{G}_2 \right) \right) \right) - y_{\mathcal{G}_3} \log \left( D \left( \phi \left( \mathcal{G}_4 \right) \right) \right) \right] \\
& + \hat{\mathbb{E}} \left[ \ell \left( f_t \left( x_t \right) \right), f_t^* \left( X_t \right) \right] + \hat{\mathbb{E}} \left[ \ell \left( f_t \left( x_g \right) \right), f_t^* \left( x_g \right) \right],
\end{aligned}
\tag{23}
$$

where $x_g$ is the generated data and $f_t^*(x_g)$ is the label of the generated data. We use the different numbers of the generated data by TOHAN (Chi et al., 2021a) and DEG-Net to train the classifier and the classification accuracy is shown in Table 6. It is clear that the performance of using data generated by TOHAN is almost the same as just using labeled data. In addition, the data generated by DEG-Net can not improve the performance of the model while the number of the target data per class is small . It may be caused by that the generated data is similar to the label target data, so that add the almost same data for the training will bring little improvement. However, it is worth nothing that the improvement will be large if the number of data generated by DEG-Net is more than 5 per class. This phenomenon indicates that that the data generated by DEG-Net is more independent to the existing target data and could be treated as the new ones in some degree.

Table 6: Classification accuracy (%) on digits FHA tasks using the generated data. Bold value represents the highest average accuracy on each column

| Task | Method | Number of Generated Data per Class | Number of Target Data per Class | | | | | | |
|------|--------|-----------------------------------|---|---|---|---|---|---|---|
| | | | 1 | 2 | 3 | 4 | 5 | 6 | 7 |
| $M \rightarrow U$ | TOHAN | 0 | **76.0** | 83.3 | 84.2 | **86.5** | **87.1** | **88.0** | **89.7** |
| | | 5 | 75.8 | **83.7** | **84.3** | 84.3 | 85.0 | 87.5 | 88.1 |
| | | 20 | 76.0 | 83.3 | 84.3 | 84.0 | 85.1 | 87.2 | 89.5 |
| | DEG-Net | 0 | **83.1** | **86.2** | **86.5** | **88.7** | 89.6 | 91.5 | 92.1 |
| | | 5 | 82.6 | 86.0 | 85.9 | 88.2 | 88.7 | 91.9 | 92.3 |
| | | 20 | 81.3 | 84.3 | 86.2 | 88.6 | **90.3** | **92.3** | **93.4** |
| $M \rightarrow S$ | TOHAN | 0 | **26.7** | **28.6** | 29.5 | **29.6** | **30.5** | 32.1 | **33.2** |
| | | 5 | 26.2 | 28.4 | 28.9 | 29.1 | 30.2 | 31.4 | 32.5 |
| | | 20 | 25.8 | 26.9 | 29.8 | 27.4 | 29.8 | **32.7** | 32.8 |
| | DEG-Net | 0 | 27.2 | **28.5** | 29.7 | **30.7** | 32.9 | 33.7 | 34.9 |
| | | 5 | **27.3** | 28.2 | 29.6 | 29.4 | 32.8 | 33.8 | 35.4 |
| | | 20 | 26.4 | 27.3 | **30.2** | 28.9 | **33.5** | **35.0** | **36.4** |

