# OpenReview forum: "Diversity of Generated Unlabeled Data Matters for Few-shot Hypothesis Adaptation"
_ICLR.cc/2023/Conference — Submitted to ICLR 2023_

### Official Review · Reviewer_1mfd · 2022-10-24

**Confidence:** 5
**Clarity, Quality, Novelty And Reproducibility:** 1. Clarity
**Correctness:** 3
**Technical Novelty And Significance:** 2
**Empirical Novelty And Significance:** 2
**Recommendation:** 3

**Strength And Weaknesses:**

### Strength
- The targeted problem of few-shot hypothesis adaptation is meaningful and challenging.

- Enhancing the diversity of generated data is reasonable and theoretically motivated.

- The paper is well-written and well-organized.

### Weakness
- The proposed method lacks novelty. The only differences compared to TOHAN are the weighted-shared generator, feature-based similarity loss, and diversity loss. The two former designs are common in previous works [1, 2]. As a result, the only novelty is the diversity loss. However, there exist very related works proposing to solve the diversity problem in image generation for better target task performance, such as [3].

- The proposed method includes many hyper-parameters ($\lambda$, $\beta$, $\gamma$). Since it is challenging to tune them since we do not have a labeled target validation set for hyper-parameter tuning in FHA or other unsupervised domain adaptation settings. I am also curious how authors determine such hyper-parameters in all adaptation tasks because the paragram "Hyper-parameter Settings" do not include the validation method.

- The empirical results do not show that "DEG-Net outperforms existing FHA methods and achieves the state-of-the art performance", as claimed by the authors before Section 2. When labeled samples are very few like 1 or 2, DEG-Net usually underperforms TOHAN, as shown in Table 2.

- The experiments only consider toy datasets like digits and CIFAR10-STL10. I understand that this paper generally follows TOHAN in the method design and experiment design. But in a very related topic called source-free domain adaptation, it is common to perform the empirical evaluation on popular domain adaptation benchmarks like Office, Office-Home, VisDA, and DomainNet. FHA is a very practical problem setting as aware by the authors, evaluation on toy benchmarks is not convincing.

- More baselines should be considered for comparison, such as recent test-time adaptation methods like Tent [4].

[1] Model Adaptation: Unsupervised Domain Adaptation without Source Data, CVPR 2020

[2] Perceptual Losses for Real-Time Style Transfer and Super-Resolution, ECCV 2016

[3] Contrastive Model Inversion for Data-Free Knowledge Distillation, IJCAI 2021

[4] Tent: Fully Test-time Adaptation by Entropy Minimization, ICLR 2021


**Summary Of The Paper:**

The paper generally follows a previous publication TOHAN (NeurIPS) to solve the few-shot hypothesis adaptation (FHA) problem. Like TOHAN, this paper proposes a method called DEG-Net consisting of two modules, i.e., the generation module and the adaptation module. The generation module generates diverse data from random noise and given the generated data and labeled target data, the adaptation module adapts the classifier to the target domain via adversarial training. The only difference between DEG-Net and TOHAN is the data generation module. DEG-Net uses a weight-shared conditional generative network instead of K (class number) generators. Besides, DEG-Net encourages the diversity of generated data by minimizing the HSIC measure of the generated data’s features. Experiments on Digits and CIFAR10-STL10 show that DEG-Net outperforms existing FHA methods in most cases.

**Summary Of The Review:**

This paper focuses on the challenging few-shot hypothesis adaptation problem and proposes to improve an existing method TOHAN by enhancing the diversity of generated data. My concern lies in the weak novelty, weak empirical evaluation, and large complexity of the proposed method.

---

### Official Review · Reviewer_7iBa · 2022-10-25

**Confidence:** 3
**Correctness:** 4
**Technical Novelty And Significance:** 3
**Empirical Novelty And Significance:** 3
**Recommendation:** 8

**Clarity, Quality, Novelty And Reproducibility:**

The paper is overall clear and well-written, with solid theoretical and empirical results. It is based on prior works, but the contribution is novel enough.

**Strength And Weaknesses:**

Strength

- The authors identify sample diversity as an important factor in auxiliary unlabeled data generation for FHA and support the finding with solid theoretical analysis.
- They further come up with practical solutions with HSIC based regularization and propose effective and non-trivial modification to TOHAN model (Chi et al).
- The empirical results are quite comprehensive and the proposed method achieves better performance in most of the cases, and the discussion on failure cases and ablation study is clear.

Weakness

- The framework of unlabeled data generation is based on existing work TOHAN, and the theoretical analysis is based on prior work related to data dependency and semi-supervised learning.
- The improvement relative to TOHAN on 6 digits FHA tasks is less significant with overlapping confidence interval.


**Summary Of The Paper:**

In this paper, the authors propose to improve the unlabeled data generation for few-shot hypothesis adaptation (FHA) which transfer a source hypothesis to target domain with only a few labeled data in target domain. They first provide theoretical proof that dependency among generated data (measured by log-coefficient) affects performance of FHA, and show that when unlabeled data is weakly dependent one can still learn a good classifier. They further propose a novel diversity-enhancing generative network that generates high diversity unlabeled while adapting the classifier to the target domain, by using HSIC. Extensive experiment results are shown on image datasets where proposed method achieves SOTA when there is good amount of target data.



**Summary Of The Review:**

The paper proposes a better unlabeled data generator for FHA problem with diversity enhancing loss. It is grounded with theoretical guarantees and achieves reasonably good empirical results.

---

### Official Review · Reviewer_TLkA · 2022-10-25

**Confidence:** 3
**Correctness:** 2
**Technical Novelty And Significance:** 2
**Empirical Novelty And Significance:** 2
**Recommendation:** 3

**Clarity, Quality, Novelty And Reproducibility:**

The paper is reasonably clear. Implementation details are included in Appendix C and should be relatively straightforward to reproduce. The novelty is relatively low and consists primarily in making the connection between the results of Dagan et al. (2019) and data-generation approaches for training few-shot classifiers.

Regarding quality: at a basic level, Theorem 1 states that if there is a large degree of dependence in the data, then a large amount of unlabeled data will be necessary to learn a good classifier. However, if I am understanding the theoretical results correctly, this degree of dependency depends on the data distribution itself rather than the synthetic data generated by a generative model. More specifically, the results of Dagan et al. (2019) assume that the marginal distributions match but there may be correlations between randomly drawn samples. This is quite a bit different from the case in FHA, where there is no guarantee that the generative model exactly matches (or is even close to) the true data distribution. From this perspective, the claim that data diversity in the generated data is important does not follow from, but is rather only loosely suggested by, the theoretical results.

Another concern is the discrepancy between the tools being used for assess dependency in the theoretical analysis, i.e. log-influence, and the regularizer used to encourage independence in the generated data, i.e. the Hilbert-Schmidt Independence Criterion. It is understandable that the log-influence is difficult to estimate, but it is unclear how, if at all, the HSIC is related to the log-influence.

A related concern is why the log-coefficient was chosen for the theoretical analysis rather than, say, Dobrushin's coefficient, which is more general.

**Details Of Ethics Concerns:**

I am concerned about the degree of overlap between this paper and that of Chi et al. (2021), which proposed TOHAN. Specifically, please compare Section 3 & 4 of this submission, starting with "Let $f^*$...", to Section 3 & 4 of Chi et al. (2021). This similarly persists down to the level of sentence structure and mathematical exposition, with minor phrase substitutions in the former case and notational substitutions in the latter. Compare also Algorithm 1 and Tables 1-2, which have substantial formatting and content overlap with Chi et al. (2021). Although Chi et al. (2021) is cited in the submission, the degree of similarity between the two works, particularly in Sections 3 and 4, struck me as rather excessive.

Chi, Haoang, Feng Liu, Wenjing Yang, Long Lan, Tongliang Liu, Bo Han, William Cheung, and James Kwok. 2021. “TOHAN: A One-Step Approach towards Few-Shot Hypothesis Adaptation.” In Advances in Neural Information Processing Systems, 34:20970–82. Curran Associates, Inc. https://proceedings.neurips.cc/paper/2021/hash/af5d5ef24881f3c3049a7b9bfe74d58b-Abstract.html.

**Strength And Weaknesses:**

Strengths:
- Investigating the effect of diversity in generated data for training few-shot classifiers is an interesting direction that should be of fairly broad interest.
- The connection between non-i.i.d. learning theoretic results (Dagan et al., 2019) and data generation in FHA is interesting.

Weaknesses:
- Both main contributions can be viewed as relatively minor extensions of TOHAN (Chi et al., 2021). On the one hand, Theorem 1 adapts the corresponding result of Chi et al. (2021) to incorporate the log-coefficient capturing dependency of the data. On the other, the methodological contribution consists primarily of adding a regularizer to TOHAN.
- The relationship to data-generation, i.e. "hallucination," approaches in few-shot learning is not discussed. At a deeper level, I did not get a strong sense why FHA was the chosen setting as opposed to standard few-shot learning (or possibly studying both), as both can rely on generated data.

**Summary Of The Paper:**

This paper considers the task of few-shot hypothesis adaptation (FHA), in which a classifier trained on a source domain is to be adapted to a target domain containing only a few labeled examples. Previous FHA approaches synthesize additional data with a generative model, but the performance boost may be limited due to lack of diversity in the generated data. The main contributions of this paper are twofold: an analysis extending previous non-i.i.d. learning theoretical results to the FHA setting thereby suggesting that the diversity of the data is important, and the proposal of a regularizer that encourages diversity in the generated examples via the Hilbert-Schmidt independence criterion.

**Summary Of The Review:**

Post-rebuttal update: I would like to thank the authors for their responses. In particular, I appreciated the clarification from the authors that the novelty lies primarily in the theoretical analysis rather than a novel algorithm. From this perspective, the two remaining concerns I have are: (1) the disconnect in assumptions between the true and synthetic data distribution, and (2) the relationship between log-influence and HSIC.

I did not see a response from the authors that addressed (1), namely that the theoretical analysis focuses on the extent to which dependency in true data distribution causes decreased generalization, but the idea behind this paper is to improve diversity in the synthetically generated data. The two may be related but it this is not clear based on the contents of the current version of the paper.

Regarding (2), the authors state that the connection between log-influence and HSIC is "obvious". I can understand that both log-influence and HSIC are both measures of independence, but if the goal is to use HSIC, then I believe that either HSIC should be used in the theoretical analysis or a more detailed discussion of the relationship between HSIC and log-influence should be included in the paper.

Overall, due to these concerns, and after having read the other reviews, I will maintain my original rating.

---

This paper investigates an interesting problem: whether data diversity in data-generation approaches to FHA is important, and what can be done to encourage such diversity. However, there are significant issues with both novelty and quality (outlined above), and therefore requires revision.

---

### Decision · Program_Chairs · 2023-01-20

**Decision:**

Reject

**Justification For Why Not Higher Score:**

The contribution is too incremental with respect to a paper published in 2021. The experimental evaluation is limited to toy datasets. The global scope appears too limited for ICLR.

**Justification For Why Not Lower Score:**

N/A

**Metareview: Summary, Strengths And Weaknesses:**

This paper studies the problem of hypothesis adaptation, also called hypothesis transfer learning, in the context of few-shot learning. They propose a generative approach to incorporate more diversity in target unlabeled data.


Strength:
-the paper addressed the important problem of diversity in few-shot learning and in the context of hypothesis adaptation which are two important problems
-the contribution has merits with theoretical and practical aspects


Weaknesses:
-the problem studied is highly dependent on a previous method called TOHAN [Chi et al.,NeurIPS 2021]
-the novelty is limited, at least in the sense that the contributions are proposed in the specific context of TOHAN and have a limited spectrum
-some aspects in the theoretical results need more justification.
-the experimental evaluation is too limited (only small toy datasets), no updates with new datasets have been proposed
-other diversity based methods in few-shot learning are not discussed.

Overall, the issue of the novelty of the work and its incremental aspect with respect to the TOHAN paper has been raised by all reviewers. In addition, the experimentions are limited to small datasets and the spectrum of the work is limited to the context of the previous paper cited. The evaluation of reviewer 7iBa highlights some merits of the paper relatively to an important problem, but the score was overestimated. There is a consensus for saying that the contribution is too incremental, the theoretical analysis still needs more justifications, the experiments are too limited, the paper must consider a larger scope about diversity in few-shot learning.
Overall, while having some merits, the contribution is not sufficient for ICLR and I propose rejection.


Other references that can be of interest:
*Mengting Chen, Yuxin Fang, Xinggang Wang, Heng Luo, Yifeng Geng, Xinyu Zhang, Chang Huang, Wenyu Liu, Bo Wang:
Diversity Transfer Network for Few-Shot Learning. AAAI 2020.
*Wang, Haoqing and Zhihong Deng. “Cross-Domain Few-Shot Classification via Adversarial Task Augmentation.” International Joint Conference on Artificial Intelligence IJCAI (2021).
*Nikita Dvornik, Julien Mairal, Cordelia Schmid: Diversity With Cooperation: Ensemble Methods for Few-Shot Classification. ICCV 2019.